# *Cryptotympana pustulata* Extract and Its Main Active Component, Oleic Acid, Inhibit Ovalbumin-Induced Allergic Airway Inflammation through Inhibition of Th2/GATA-3 and Interleukin-17/RORγt Signaling Pathways in Asthmatic Mice

**DOI:** 10.3390/molecules26071854

**Published:** 2021-03-25

**Authors:** Seung-Hyung Kim, Jung-Hee Hong, Won-Kyung Yang, Hyo-Jung Kim, Hyo-Jin An, Young-Cheol Lee

**Affiliations:** 1Institute of Traditional Medicine & Bioscience, Daejeon University, Daejeon 34520, Korea; sksh518@dju.ac.kr (S.-H.K.); ywks1220@dju.ac.kr (W.-K.Y.); 2Department of Herbology, College of Korean Medicine, Sangji University, 83 Sangjidae-gil, Wonju 26339, Korea; anifam@hanmail.net; 3Department of Pharmacology, College of Korean Medicine, Sangji University, 83 Sangjidae-gil, Wonju 26339, Korea; hyojung_95@naver.com (H.-J.K.); hjan@sangji.ac.kr (H.-J.A.)

**Keywords:** Cicadae Periostracum, asthma, IL-5, IL-13, GATA-3, RORγt

## Abstract

Cicadae Periostracum (CP), derived from the slough of *Cryptotympana pustulata*, has been used as traditional medicine in Korea and China because of its diaphoretic, antipyretic, anti-inflammatory, antioxidant, and antianaphylactic activities. The major bioactive compounds include oleic acid (OA), palmitic acid, and linoleic acid. However, the precise therapeutic mechanisms underlying its action in asthma remain unclear. The objective of this study was to determine the antiasthmatic effects of CP in an ovalbumin (OVA)-induced asthmatic mouse model. CP and OA inhibited the inflammatory cell infiltration, airway hyperresponsiveness (AHR), and production of interleukin (IL)7 and Th2 cytokines (IL-5) in the bronchoalveolar lavage fluid and OVA-specific imunoglobin E (IgE) in the serum. The gene expression of IL-5, IL-13, CCR3, MUC5AC, and COX-2 was attenuated in lung tissues. CP and OA might inhibit the nuclear translocation of GATA-binding protein 3 (GATA-3) and retinoic acid receptor-related orphan receptor γt (RORγt) via the upregulation of forkhead box p3 (Foxp3), thereby preventing the activation of GATA-3 and RORγt. In the in vitro experiment, a similar result was observed for Th2 and GATA-3. These results suggest that CP has the potential for the treatment of asthma via the inhibition of the GATA-3/Th2 and IL-17/RORγt signaling pathways.

## 1. Introduction

Cicadae Periostracum (CP) has been used in traditional medicine in Korea and China to relieve pathogenic wind-heat from the lung channel and in the treatment of hoarseness, sore throat, spasms, itching, and other symptoms [1]. It is the most commonly used herbal medicine, possessing diverse pharmacological activities such as anti-inflammatory, antioxidant [2,3], anticonvulsive, hypothermic, sedative [2], antiallergic, antipyretic [4], and antibacterial [5] activities. Recently, a regulatory compound for Th17 cell differentiation was isolated from CP, which showed inhibitory effects on Th17 differentiation [6]. Recent studies have shown that CP is the most frequently prescribed single-herb medicine in allergic rhinitis [7]. The aqueous extract of CP significantly reduced compound 48/80-induced systemic anaphylactic reactions [8].

Recently, it was shown that the CP contains various kinds of ingredients, many fatty acids, amino acids, protein, chitin, and acetyldopamine dimers. Oleic acid (OA) exhibits anti-inflammatory activity, and the most abundant unsaturated fatty acid in CP is OA, followed by linoleic acid [9]. Other fatty acids were detected in small quantities. Several different types of amino acids, such as aspartic acid, tyrosine, and valine, are present in the CP [10]. Fatty acids, including omega-3 and -6, play key roles in the development of numerous inflammatory mediators involved in the pathophysiology of allergic asthma [11]. It was reported that the high dietary intake of OA is positively associated with asthma in adults [12]. Although some controversial results have been reported, OA is an anti-inflammatory fatty acid that plays an essential role in the activation of different pathways of immune cells [13]. Therefore, OA was selected as one of the major bioactive ingredients of the CP. The CP can be used as a useful anti-inflammatory traditional herbal medicine, and we hypothesized that CP could reduce allergic airway inflammation.

Asthma is a complex chronic airway disease characterized by Th2-predominant eosinophilic lung inflammation, variable airflow limitation, airway hyperresponsiveness (AHR), remodeling, epithelial cell hyperplasia, airway wall thickening, goblet cell hyperplasia, mucus hypersecretion, and subepithelial fibrosis [14,15]. Activated Th2 cytokines induce eosinophil infiltration, and Th2-type asthma with eosinophilia is a common phenotype in asthma. Th2 cytokines, such as interleukin-4 (IL-4), IL-5, and IL-13, can initiate and sustain important pathophysiological features of asthma [16]. IL-4 is essential for the allergic response and immunoglobin E (IgE) production, and IL-5 is important for eosinophil survival and expansion, which contribute to lung injury in asthma. IL-13 has pleiotropic functions in the lungs, including playing a crucial role in the development of AHR and airway remodeling [17]. Therefore, efforts to alter the Th1–Th2 balance in asthma are strategically required, either by suppression of Th2 cytokines or upregulation of the Th1 response. Several transcription factors, such as GATA-binding protein 3 (GATA-3), c-maf, NFATc1, and STAT6, have been found to increase Th2 cytokine production, although GATA-3 is expressed selectively in Th2 cells. In particular, the GATA-3 transcription factor is known to induce the expression of Th2 cytokines, including IL-4, IL-5, and IL-13 [18]. Th17 cytokine (IL-17) mediates the recruitment of neutrophils and eosinophils, and can activate epithelial cells and airway smooth muscle cells to produce proinflammatory mediators. Th17 cells can upregulate Th2-cell-mediated eosinophilic airway inflammation [19]. Th17 cells express IL-17 and the essential transcription factor retinoic acid receptor-related orphan receptor γt (RORγt). IL-17A levels are upregulated in the lung tissues of patients with asthma, and the level of IL-17 correlates with asthma severity in patients with neutrophilic asthma [20]. Th17 cells also aggravate Th2-cell-mediated eosinophilic airway and lung inflammation [21]. Forkhead box p3 (Foxp3+) regulatory T cells (Treg cells), which have been shown to play an important role in asthma, and can suppress effector CD4+ T cell responses. In addition, Foxp3+ Treg cells can attenuate Th2- and Th17-cell-mediated airway inflammation and hyperresponsiveness both in animal models and in patients with asthma [22]. The downregulation of Th17 cell expansion is associated with the mutual regulation of Foxp3+ Treg cells. RORγt transcription factor upregulation and Foxp3 downregulation were correlated with the loss of Foxp3+ Treg cell function [23]. Therefore, in addition to Th2 cells, Th17 cells, RORγt, and Foxp3+ Treg cells may be important therapeutic targets in asthma and airway inflammation.

Despite their numerous pharmacological activities, the therapeutic mechanisms of the action of the CP and OA have not been clearly elucidated. In addition, there are no reports on the antiasthmatic effects of CP in vivo. In this study, we investigated the antiasthmatic effects of the CP and the involvement of the Th2/GATA-3 and IL-17/RORγt pathways in a mouse model of asthma.

## 2. Results

### 2.1. Chemical Profiles of Fatty Acids in CP Ethanol Extract

The main chemicals of the CP were found to be crude fat, which comprised 0.62% of the CP. As shown in Table 1, approximately 17 compounds were identified in the ethanol extract of the CP using a gas chromatography-flame ionization detection (GC-FID). OA, palmitic acid, stearic acid, and linoleic acid were the main components of the ethanol extract of the CP. The relative contents of the compounds, expressed as g/100 g, are listed in Table 1. Among these, OA, palmitic acid, stearic acid, and linoleic acid were the major fatty acids. 

The GC-FID analysis results showed the peak profiles of several compounds, and the eluent was detected at 220 nm. In addition, some characteristic peaks were found in the GC peak profiles; however, the chemicals responsible for the peaks were not identified in this study. 

### 2.2. Inhibitory Effects of OA, CP, and Dexamethasone on Ovalbumin (OVA)-Induced AHR in Mice and Inflammatory Cell Infiltration into the Lung and Bronchoalveolar Lavage Fluid (BALF)

We determined the inhibitory effects of OA and CP on AHR in vivo using the methacholine test (enhanced pause (Penh) system), which uses a gradational increase in doses of methacholine to determine whether OA and CP improved the AHR in a mouse model of allergic asthma. AHR induced by methacholine at concentrations of 3.125, 6.25, 12.5, and 25 mg/mL was assessed on the first day after the final OVA challenge. As shown in Figure 1B, the Penh values were significantly higher in OVA-induced control mice than in normal mice. We found that at a high inhaled dose of methacholine (25 mg/mL) in OA- (10 and 20 mg/kg), CP- (100 and 200 mg/kg), and dexamethasone-treated mice with asthma, Penh values decreased in a dose-dependent manner compared with the control group sensitized with OVA. OA and CP significantly inhibited AHR in asthmatic mice. To assess the antiasthmatic effects of OA and CP in a mouse model of allergic asthma, the total number of BALF and lung cells and the number of neutrophils/eosinophils were detected in the BALF and lung tissues. As shown in Figure 1, the total number of BALF cells isolated from normal mice was (12.3 ± 2.61) × 10^4^ cells/mL, whereas a significantly higher number, (119.0 ± 20.23) × 10^4^ cells/mL, was isolated from the OVA-indued control mice. OA (20 mg/kg), CP (200 mg/kg), and dexamethasone reduced the OVA-induced recruitment of total lymphocytes into the BALF (Figure 1C). A significant increase in inflammatory cell infiltration to the lung tissues was observed in the OVA-challenged mice. OA (20 mg/kg), CP (100, 200 mg/kg), and dexamethasone reduced the total number of lung cells and the number of neutrophils and eosinophils in BALF (Figure 1D–F).

### 2.3. Histological Analysis of Lung Tissues

To investigate the inhibitory effects of OA and CP on the histological changes in the OVA-induced asthmatic mice, the lung tissues were stained with hematoxylin and eosin (H&E), Masson’s trichrome (M-T), and periodic acid-Schiff (PAS), and the trachea was stained with Alcian Blue PAS (AB-PAS) staining solution. H&E staining of the lungs showed that OA and CP reduced the infiltration of various inflammatory cells, including eosinophils, neutrophils, mast cells, and lymphocytes, in the peribronchiolar and perivascular regions of the lungs in asthmatic mice, similar to the results obtained after dexamethasone treatment (Figure 2). M-T staining showed that subepithelial fibrosis and collagen deposition in the peribronchial and perivascular tissues was lower in OA- and CP-treated mice than in untreated mice. We used PAS and AB-PAS staining to investigate mucus hypersecretion and goblet cell hyperplasia in the bronchus, which demonstrated that OA, CP, and dexamethasone reduced goblet cell hyperplasia compared with OVA-sensitized mice (Figure 2C,D). The number of PAS and AB-PAS-positive goblet cells was significantly higher in the OVA-induced control group than in the normal group. Treatment with OA and CP resulted in a significantly lower number of PAS-or AB-PAS-stained cells. Taken together, OA and CP suppressed the histopathological alterations in the lungs and tracheal tissue of asthmatic mice.

### 2.4. Suppressive Effects of OA, CP, and Dexamethasone on Th2 Cytokines in BALF and Spleen and IgE Production in Serum

In asthmatic mouse models, Th2 cytokines (IL-4, IL-5, and IL-13) and IL-17A are important for eosinophil and neutrophil infiltration and OVA-specific IgE production [24]. To further investigate whether OA, CP, and dexamethasone affected the production of Th2 cytokines and IL-17 in BALF, the cytokine levels of IL-4, IL-5, IL-13, and IL-17 were assessed by ELISA after the final OVA challenge. As shown in Figure 3, IL-5 and IL-17 levels were significantly lower in the CP- (100 and 200 mg/kg), OA- (20 mg/kg), and dexamethasone-treated mice than in OVA-induced control mice. However, there were no significant differences in IL-4, IL-13, and interferon (IFN)-γ production in BALF (data not shown). IFN-γ, a representative Th1-type cytokine, was not detected at a significant level in BALF from any of the mice groups, which is consistent with the results obtained in other investigations using a mouse model of allergic asthma [25]. 

As the OVA-specific IgE level is a crucial aspect of the OVA-induced asthmatic mouse model, we investigated the levels of OVA-specific IgE in the serum. As shown in Figure 3B, the levels of OVA-specific IgE in the sera of control mice were higher than those in sera from normal mice. OA (20 mg/kg), CP (100 and 200 mg/kg), and dexamethasone significantly suppressed OVA-specific IgE levels. In particular, CP reduced the levels of OVA-specific IgE in a dose-dependent manner. However, OA (10 mg/kg) weakly inhibited OVA-specific IgE in the serum without significant suppression. Consistent with the above results, the assessment of Th2 cytokine levels (IL-4, IL-5, and IL-13) in culture supernatants by ELISA showed that OA, CP, and dexamethasone inhibited the production of these cytokines (Figure 3C). In contrast, CP (200 mg/kg) treatment markedly increased the levels of the Th1 cytokine (IFN-γ). These results indicate that CP may regulate the Th1/Th2 balance. 

### 2.5. Effects of OA, CP, and Dexamethasone on IL-5, IL-13, IL-4, CCR3, MUC5AC, and COX-2 mRNA Expression in Lung Tissues

As shown in Figure 3, since CP and OA inhibited IL-5 in BALF and Th2 cytokine (IL-5, IL-4, and IL-13) production in spleen cells, we assessed the mRNA expression of asthma-associated cytokines and mediators in the lung tissues of OVA-induced asthmatic mice using qRT-PCR (Figure 4). Additionally, we investigated whether the patterns of gene expression of Th2 cytokines and others were significantly correlated with protein expression after treatment with CP and OA. Similar to the results of protein expression, treatment with CP and OA inhibited the mRNA expression of IL-5. The mRNA expression of IL-13 was suppressed by CP and OA. The Th2-cell-mediated responses result in the secretion of a subset of chemokines and chemokine receptors. Inflammatory cell recruitment in asthma consists primarily of Th2 cells, neutrophils, mast cells, and eosinophils. The CC chemokine receptor CCR3 is thought to be specific to eosinophils and Th2 cells [26]. However, CP and OA did not inhibit IL-17 mRNA levels in the lungs (data not shown). It is unclear why IL-17 mRNA expression was not significantly suppressed. A possible explanation is that various transcription factors, such as RORα, RORγT, signal transducer and activator of transcription (STAT) 3, nuclear factor-κB (NF-κB), and activator protein (AP)-1, have been shown to induce Th17 cytokine production [15].

We investigated the mechanism through which OA and CP attenuated eosinophil recruitment into the lungs through the analysis of the mRNA expression of CCR3 in lung tissues. OA and CP (200 mg/kg) significantly inhibited the mRNA expression of CCR3 (Figure 4D). MUC5AC, a specific marker of goblet cells, is the major respiratory mucin secreted by the submucosal glands and bronchial epithelium [14,27]. It may cause airway mucus hypersecretion. Products of the COX-2 pathway, such as prostaglandins, increase Th17 cell differentiation of naive CD4+ T cells during allergic airway inflammation in mice [28]. Remarkably, our results showed that CP and OA significantly suppressed the mRNA expression of MUC5AC and COX-2 (Figure 4E,F). The above results are correlated with the changes in eosinophil recruitment (Figure 1 and Figure 2, Table 2), BALF cytokines (IL-5 and IL-17A) (Figure 3), and OVA-specific IgE levels in serum (Figure 3B).

### 2.6. Suppressive Effects of OA, CP, and Dexamethasone on CD11b + Gr-1 + (High) Neutrophils and SiglecF + CD11b+ Eosinophil Infiltration in the Lung Tissue and BALF of Asthmatic Mouse Model

Although most asthmatic mice models present with eosinophilic airway inflammation, several asthmatic mice models with severe asthma show substantial neutrophil recruitment to airway and lung tissues [14]. Gr-1 comprises two components: Ly6G and Ly6C. Ly6G is exclusively expressed in neutrophils [29], and CD11b is highly expressed in eosinophils [30]. CD11b + Gr-1+ cells include monocytes, neutrophils, and eosinophils. The CD11b + Gr-1 + (high) population may constitute a substantial portion of neutrophils [31]. SiglecF is mostly expressed in mouse eosinophils [32]. Suppressive effects of OA, CP, and dexamethasone on CD11b + Gr-1 + (high) neutrophils and SiglecF + CD11b+ eosinophils were investigated by flow cytometry analysis in the lungs and BALF of the asthmatic mouse model. The proportion and the absolute number of CD11b + Gr-1 + (high) neutrophils in the BALF and lung cells of OVA-induced asthmatic mice were higher than those in the normal control group, whereas the proportion and the absolute number of these cells in mice treated with OA, CP, and dexamethasone were lower than those in the OVA-induced control mice (Table 2). The expansion of CD11b + Gr-1 + (high) neutrophils and SiglecF + CD11b+ eosinophils in the BALF and lung cells of the OVA-induced control mice correlated with an increase in IL-5, IL-13, CCR3, and IL-17A in the BALF and lung (Figure 3 and Figure 4). Our results revealed that OA and CP significantly decreased the number of CD11b + Gr-1 + (high) neutrophils and SiglecF + CD11b+ eosinophils (Table 2). The reduction in the number of CD11b + Gr-1+ neutrophils and SiglecF + CD11b+ eosinophils was correlated with reductions in the numbers of neutrophils and eosinophils in BALF, total number of BALF cells, and total number of lung cells (Figure 1C–F).

### 2.7. Immunofluorescence Staining of GATA-3, RORγt, and Foxp3 Transcription Factors in the Lung of Asthmatic Mouse Model

The GATA3 transcription factor is important for the differentiation of naïve T cells into Th2 cells, and it promotes the secretion of Th2 cytokines, including IL-4, IL-5, and IL-13 [18]. Foxp3+ regulatory T cells (Tregs) play an anti-inflammatory role and have an inhibitory relationship with Th17 cells. RORγt has been identified as a master regulator of Th17 cell differentiation. The balance between Foxp3+ Tregs and Th17 cells is crucial in autoimmune diseases, including asthma. Foxp3+ Tregs can suppress Th17 cell differentiation by inhibiting RORγt activity [33]. To confirm the expression of these transcription factors, we measured the expression of GATA-3, RORγt, and Foxp3 transcription factors in the lung tissue of asthmatic mice.

As shown in Figure 5, exposure to OVA resulted in the upregulation of GATA-3 and RORγt protein expression and downregulation of Foxp3 expression in mice, suggesting that OVA might induce asthma through the GATA-3, RORγt, and Foxp3 signaling pathways. By contrast, high concentrations of CP and OA suppressed the level of the GATA-3 protein and enhanced the level of the Foxp3 protein (Figure 5). Additionally, treatment with CP and OA (low and high dosages) inhibited RORγt protein expression. These results indicate that CP and OA exert a therapeutic effect against asthma, possibly via the GATA-3, RORγt, and Foxp3 pathways.

### 2.8. Suppressive Effect of CP, OA, and Dexamethasone on Cytokine Expression In Vitro (EL-4 and LA-4 Cells) 

EL-4 and LA-4 cells were treated with CP and OA at various concentrations, but there was no influence on cell viability (assessed by the MTT assay, Appendix A). To confirm the possible effect of CP and OA on Th2 cytokine, TNF-α, and GATA-3 expression, we tested the mRNA expression levels of IL-4, IL-5, IL-13, TNF-α, and GATA-3 in phorbol 12-myristate 13-acetate (PMA)- and 4-tertoctylphenol (OP)-stimulated EL-4 T cells. Additionally, we investigated the mRNA expression levels of CCL2 and CCL5 in lipopolysaccharide (LPS)-stimulated LA-4 cells. As shown in Figure 6, the expressions of IL-4, IL-5, IL-13, TNF-α, and GATA-3 in the media increased by stimulation with PMA/OP. When EL-4 T cells were treated with CP (100 or 200 μg/mL) and OA (100 or 300 µM), their secretion induced by PMA/OP significantly reduced (Figure 6). Previously, CCL2/monocyte chemotactic protein-1 (MCP-1) was found to have higher levels of CCL2 and CCL5 (regulated on activation, normal T cells expressed, and secreted (RANTES)) in the BALF, associated with the development of status asthmaticus, along with increased IL-5, compared with patients with mild asthma [34].

Treatment with CP and OA in LA-4 cells inhibited the expression of CCL2 and CCL5, the chemokines that recruit monocytes and eosinophils to inflammatory tissues (Figure 6F,G), whereas the CCL2 levels in LA-4 cells were not decreased by treatment with OA (low dosage) compared with the control group. These results suggest that the inhibition of Th2 cytokines by CP and OA may be mediated through a reduction of expression of the GATA-3 transcription factor in EL-4 T cells. These results correlate with the results of Th2 cytokine and GATA-3 expression in lung tissues (Figure 3, Figure 4 and Figure 5).

## 3. Discussion

Asthma is a Th2-cell-driven inflammatory disease, characterized by Th2 cytokine production, AHR, and eosinophil and neutrophil infiltration [14]. Numerous cell types, including T lymphocytes, eosinophils, neutrophils, epithelial cells, and mast cells, have been implicated in the pathogenesis of asthma. In particular, Th2 cells play a crucial role in asthma pathogenesis through the production of chemokines and cytokines. Airway remodeling, including subepithelial collagen deposition, goblet cell hyperplasia, airway smooth muscle hypertrophy, and mucus hypersecretion, are also prevalent [15,35]. Animal models of asthma are mostly induced by the administration of allergens to the airway and lungs via aerosolization and intraperitoneal injections of an allergen (commonly OVA) in conjunction with aluminum hydroxide as an adjuvant [14]. Therefore, we used a murine model of OVA-induced allergic asthma, as shown in Figure 1A. 

As previously described in the Introduction section, CP has a range of pharmacological activities, including antiallergic, antioxidant, and anti-inflammatory activities. However, the chemical compounds and antiasthmatic mechanisms of CP have not been studied in depth. Therefore, a chemical profile analysis was performed to determine the major constituents of the CP extract and identify the main chemical components through fatty acid analyses. As shown in the results (Table 1), the main compounds of CP are fatty acids, present at concentrations of 0.62 g/100 g. In total, as shown in Table 1, 17 compounds were identified in the ethanol extract of CP using GC-FID. OA, palmitic acid, stearic acid, linoleic acid were the main chemicals in the ethanol extract of CP. 

Recent studies have shown that the CP has inhibitory effects on Th17 differentiation [6], allergic rhinitis [7], anaphylactic reactions [8], and contact dermatitis [36]. The CP improved kidney inflammation and fibrosis in IgA nephropathy rat models by reducing MCP-1, TNF-α, IL-1β, and IL-6 levels [37]. Chang et al. [38] reported that the CP inhibited oxidative stress and inflammation in keratinocytes. OA, which is the major component of CP, reduced IgE binding to the allergens [39]; *Camellia japonica* oil, which contains OA as the major bioactive component, suppressed asthma incidence through the GATA-3/Th2 cytokine pathways [40]. In vitro studies have reported that treatment with oleate (the salts and esters of OA) inhibited E-selectin, vascular cell adhesion molecule-1 (VCAM-1), and intercellular adhesion molecule-1 (ICAM-1) expression in endothelial cells [41]. Verlengia et al. reported a reduction in the production of IFN-γ and IL-2 and a suppressive effect of OA on the proliferation of Jurkat T cells [42]. OA was reported to modulate colitis by reducing IL-8 synthesis and oxidative stress [43]. Additionally, OA performs anti-inflammatory functions by increasing the level of the anti-inflammatory cytokine IL-10 and reducing the levels of IL-6 and TNF-α [44]. 

Likewise, CP and OA could be considered antiasthmatic drug candidates. However, the mechanisms through which CP and OA exert their effects are still unclear. Further studies are needed to evaluate the precise mechanism through which CP inhibits airway inflammation in mice models of asthma. These preliminary results indicate that CP has the potential to be used in the treatment of asthma. Therefore, we hypothesized that CP may suppress airway inflammation in a mouse model of asthma. 

Our results showed that both CP and OA suppressed the levels of Penh, total lymphocytes, neutrophil and eosinophil infiltration, subepithelial fibrosis, collagen deposition, goblet cell hyperplasia, mucus secretion, and lung inflammation; and decreased the levels of Th2 cytokines (IL-4, IL-5, and IL-13) and IL-17A in BALF and OVA-specific IgE in sera (Figure 2, Figure 3 and Figure 4). The mRNA expression levels of IL-5, IL-13, CCR3, MUC5AC, and COX2 in lung tissues were reduced in CP- and OA-treated mice. Dexamethasone was shown to improve allergic lung inflammation by decreasing eosinophil counts and Th2 cytokine production through the inhibition of NF-kB [45]. Moreover, dexamethasone has a well-known clinical effect on the improvement in lung function in asthma; therefore, we used dexamethasone as a positive control. 

As described in the Results section, CD11b + Gr-1+ cells include neutrophils, eosinophils, and monocytes. The CD11b + Gr-1 + (high) population may constitute a substantial portion of neutrophils and eosinophils [31,46]. SiglecF is mostly expressed in mouse eosinophils [32]. Our results showed that OA, CP, and dexamethasone significantly inhibited the absolute numbers of these cells in mice (Table 2). These results correlated with the decreases in the IL-5, IL-13, CCR3, and IL-17A levels in the BALF and lungs (Figure 3 and Figure 4); reduction in the number of neutrophils/eosinophils in the BALF; total number of BALF cells; and total number of lung cells (Table 2).

Th2 cytokines, including IL-4, IL-5, and IL-13, are regulated by the transcription factor GATA-3 [47], which mediates allergic airway inflammation. IL-17A plays a key role in neutrophil response in the lung; IL-17A is involved in the pathogenesis of allergic asthma and airway remodeling [48] and induces neutrophil infiltration [49]. RORγt plays an important role in Th17 cell differentiation and is required for IL-17A secretion [50]. Foxp3 drives Treg differentiation and inhibits RORγt function [51]. When immune cells are stimulated by proinflammatory cytokines, Foxp3 function is inhibited and Th17 differentiation is induced [51]. Foxp3+ Treg cells can inhibit Th17- and Th- cell-mediated inflammatory responses and prevent bronchial AHR and airway inflammation both in animal models and in patients with asthma; the functions of Foxp3+ Treg cells are impaired in asthma [52]. As shown in Figure 5, high concentrations of CP and OA suppressed the level of GATA-3 protein and enhanced the level of Foxp3 protein (Figure 5). Treatment with CP and OA (low and high dosages) suppressed RORγt protein expression. These results showed that CP and OA exerted a therapeutic effect against asthma, possibly via the GATA-3, RORγt, and Foxp3 pathways. 

IL-5 and IL-13 showed strong correlations with AHR and CCL2 (MCP-1) in asthma severity and fast lung function decline [53]. Additionally, CCL5 (RANTES) has been identified in the airways of patients with asthma [26,34]. High levels of CCL2 and CCL5 in the BALF along with increased IL-5 in patients with asthma induce eosinophil infiltration and activation [54]. The CCR3 receptor, which is highly expressed on eosinophils and differentially expressed on Th2 cells, is involved in the activation and degranulation of eosinophils [26]. 

As shown in Figure 6, treatment with CP and OA in LA4 cells suppressed the expression of CCL2 and CCL5, the chemokines that recruit monocytes and eosinophils to inflammatory tissues (Figure 6F,G), whereas CCL2 levels in LA-4 cells were not decreased by treatment with OA (low dosage) compared with the control group. These results suggest that the inhibition of Th2 cytokines by CP and OA may be mediated through reduced expression of the GATA-3 transcription factor in EL-4 T cells. These results corroborate the results for Th2 cytokines and GATA-3 in lung tissues (Figure 3, Figure 4 and Figure 5).

Our results showed that CP and OA exerted suppressive effects on airway and lung inflammation and that these effects were caused by the inhibition of Th2 cytokines, IL-17, and OVA-specific IgE through the GATA-3, RORγt, and Foxp3 transcription pathways. 

The CP is a commonly used crude drug in traditional medicine. Based on preliminary studies [1,8,37,55,56,57,58] and including our supplementary data (Appendix A), which showed the safety of CP and OA, safe and effective concentrations were used in the current study. Thus, our results indicate that CP and OA have the potential for use in the treatment of patients with allergic asthma.

## 4. Materials and Methods

### 4.1. Preparation of Crude Extracts of CP and Chemical Reagents

The CP samples were purchased from Onggi Korean Herbal Medicine Market (Daegu, Korea) figurein April 2019, and herbal identification was performed by Professor Young-Cheol Lee, College of Korean Medicine, Sangji University, Wonju, Korea. A voucher specimen (no. 2019-SJCP-1) was deposited in the laboratory of the Department of Herbology, College of Korean Medicine, Sangji University Wonju 26339, Republic of Korea. Dried and chopped CP (600 g) was extracted three times with 70% ethanol using a 3 h reflux. The materials were filtered under reduced pressure at 40 °C using a vacuum rotatory evaporator (BUCHI B-480, Buchi, Flawil, Switzerland) and dried in a freeze-drier (EYELA FDU-540, Japan) to yield the CP extract (29.56 g). The yield (*w/w*) of the extract was approximately 4.93%. Dexamethasone (Sigma Aldrich, Korea), which was dissolved in saline, was used as a positive control. Methacholine, OVA, OA, and aluminum hydroxide were purchased from Sigma-Aldrich. All other chemicals and solvents used in the experiments were of analytical grade and were purchased from Sigma-Aldrich, unless otherwise indicated.

### 4.2. Fatty Acid Composition and Total Crude Fat Analysis 

The fatty acid composition and the total fat content of CP were analyzed using a previously described method [59,60]. Briefly, the CP sample was subjected to transmethylation/methylation under sequential alkaline and acidic conditions for fatty acid identification and quantitation by gas chromatography-flame ionization detection (GC-FID). Fatty acid methyl esters (FAMEs) were analyzed using capillary gas chromatography, as previously described [59,61]. The resulting peaks were identified by comparison with a certified standard material (FAME mix, C4:0 to C24:0, purchased from Sigma Aldrich Korea, Seoul, Korea) and quantified by area normalization. The total crude fat content of the CP was analyzed using diethyl ether by the Soxtec method and calculated from the total amount of the individual fatty acid composition. 

### 4.3. Animals 

Seven-week-old male BALB/c mice, weighing 20–25 g, were obtained from OrientBio Co. Ltd. (Seongnam, Republic of Korea) and housed at 21 ± 2 °C, under 50% ± 5% relative humidity and a 12 h light/dark cycle. They had free access to a commercial diet and water and were maintained in specific pathogen-free conditions at our animal breeding facilities. All animal experiments and procedures were approved by the Committee for Animal Welfare at Daejeon University, Republic of Korea (DJUARB2019-027) on 1 October 2019. This study was carried out in accordance with the Guidelines of the Institutional Animal Care and Use Committee of the South Korea Research Institute of Bioscience and Biotechnology (Daejeon, Republic of Korea).

### 4.4. OVA Allergen Sensitization, Challenge, and Enhanced Pause (Penh) Measurement

OVA preparation, sensitization, challenge, and Penh measurements were performed according to a method previously described [61,62] with minor modifications. Briefly, the mice were immunized by intraperitoneal (i.p.) injections of 200 µL of alum-precipitated antigen, containing 12.5 μg OVA (Sigma Aldrich, St. Louis. MO, USA) and 0.26 mg aluminum hydroxide in phosphate-buffered saline (PBS), 14 days after the first sensitization. They were then administered intratracheal injections of 100 µL of 2 mg/mL OVA in PBS 10 days after the first sensitization. On day 14, on the back of the tongue, the mice were exposed to 1% OVA in saline aerosolized with the ME-U12 ultrasonic nebulizer (Omron, Tokyo, Japan) for 30 min/day, 3 days/week, from days 14 to 35. Then, the mice were exposed to inhalation of 2% aerosolized OVA in saline from days 35 to 42, as previously described in Figure 1A. CP (100 and 200 mg/kg), OA (10 and 20 mg/kg), and dexamethasone (Dexa, 3 mg/kg) were orally administered three times per week for approximately 4 weeks (from days 21 to 44). One day after the final OVA exposure, Penh values were measured, and BALF, lung cells, lung tissues, and serum were collected for molecular analyses. The mice were separated into seven groups (n = 10 per group) as follows: (1) Normal control, (2) OVA-control, (3) OVA-dexamethasone (Dexa.) at 3 mg/kg (i.p.), (4) OVA-OA 10 mg/kg, (5) OVA-OA 20 mg/kg, (6) OVA-CP 100 mg/kg, and (7) OVA-CP 200 mg/kg. Dexamethasone was used as a positive control. The experimental scheme is presented in Figure 1A.

The Penh value, the airway’s responsiveness to methacholine, is equal to Pause × PEF/PIF, where PIF is the peak inspiratory flow, PEF is the peak expiratory flow, Pause = (Te–Tr)/Tr, Te is the expiratory time, and Tr is the relaxation time. At 24 h after the final inhalation, the mice were placed in a plethysmographic chamber, and aerosolized methacholine at increasing concentrations (3.125, 6.25, 12.5, or 25 mg/mL) was nebulized through the inlet of the chamber. Airway responsiveness was then monitored for 30 min and is expressed as an enhanced pause that correlates with airway resistance. The differences in Penh values among the groups were analyzed using an unpaired Student’s *t*-test.

### 4.5. BALF 

According to a modified method previously described [62,63], after Penh measurement, the mice were anesthetized by i.p. injection of 0.2 mL of sodium pentobarbitone (60 mg/kg body weight) in PBS. Tracheotomies were performed on the mice, and BALF was obtained by lavaging the lung and airway lamina with saline via tracheal cannulation. Briefly, the BALF cells in the lungs were recovered by flushing the lung three times with 10% fetal bovine serum (FBS) and 1 mL of PBS with 1 mM ethylenediaminetetraacetic acid (EDTA) through the trachea and centrifuged at 400× *g* for 5 min at 4 °C. The total cell number was counted using a hemocytometer, and the slides were prepared using a cytospin apparatus (Cellspin, Hanil Science, Korea). Then, 100 µL of fluid with 2–4 × 10^3^ cells was transferred to cytospin slides using a cytospin centrifuge (400× *g*, 4 min). Differential cell counts in BALF were performed in accordance with standard morphological criteria after staining with a Diff-Quik Stain Set (Baxter Healthcare Corp., Miami, FL, USA) by two independent investigators. The remaining BALF supernatants were stored at −25 °C until further use.

### 4.6. Collection of Lung Tissues and Lung Cell Preparation

Single-cell suspensions from lung tissues were isolated by mechanical disruption and collagenase digestion using DMEM (Dulbecco’s Modified Eagle’s medium) supplemented with 50 μM 2-mercaptoethanol, 2 mM l-glutamine, 20 mM HEPES (Hydroxyethyl piperazine Ethane Sulfonic acid), 100 U/mL penicillin, 100 µg/mL streptomycin, and 2% heat-inactivated FBS. Briefly, the isolated right lobes of the lungs were removed from the thoracic cavity and minced using sterile scalpels, followed by incubation in PBS containing 2 mg/mL dispase and 1 mg/mL collagenase IV for 30 min at 37 °C in a sterile polypropylene tube. After incubation for 30 min, the lung tissues were pipetted vigorously to dissolve the remaining tissue clumps and then filtered through a 70 μm cell strainer. The total number of lung cells was determined using a hemocytometer. The remaining lobes of the left lung were stored for histological analyses (bottom lobe) and mRNA extraction (upper lobe).

### 4.7. H&E, M-T, PAS, and AB-PAS Staining

For histopathological analysis, harvested lung sections were inspected by morphological evaluations as described previously with minor modifications [63,64]. Briefly, the lung sections were fixed with 4% paraformaldehyde, embedded in paraffin, and cut into 3 µm sections. The lung sections were stained with H&E for general morphology, and the inflammatory lymphocytic cell infiltration into lung tissues was measured by optical microscopic evaluation of the H&E-stained sections. Other sections were stained with M-T to determine subepithelial fibrosis and collagen deposition, and with PAS or AB/PAS to quantify mucus secretion and goblet cell hyperplasia in the lung epithelium. The quantitation of the peribronchial or perivascular inflammation score was performed as described previously [64] with modifications. 

### 4.8. Fluorescence-ActivatedCellSorting. (FACS) Analysis

All antibodies, such as SiglecF, anti-cluster of differentiation (CD)11b, and Gr-1 for FACS, were purchased from Becton Dickinson (BD) PharMingen (San Diego, CA, USA). Briefly, BALF and lung cells (5 × 10^5^) were washed with FACS staining buffer. Then, they were stained with the indicated antibodies in FACS staining buffer (PBS containing 1% FBS and 0.01% NaN_3_) for 10 min on ice. Subsequently, the stained cells were analyzed by two-color flow cytometry using an FACSCalibur device, and data analysis was performed using CellQuest software (BD Biosciences, Mountain View, CA, USA) as previously described [61,63].

### 4.9. Enzyme-Linked Immunosorbent Assay (ELISA) 

IL-5 and IL-17A production in BALF and anti-OVA-specific IgE levels in serum were measured using monoclonal antibody-based mouse interleukin ELISA kits, following the manufacturer’s instructions (R&D Systems, Minneapolis, MN, USA). Spleen cells isolated from individuals within each group were pooled before performing the experiments. Spleen cells (1 × 10^5^ cells/well) were stimulated with or without 1 μg/mL OVA for 48 h at 5% CO_2_, 37 °C, and 90% humidity in 96-well culture plates (Corning, Cambridge, MA, USA). The spleen cell culture supernatants were collected, and cytokine production in the culture supernatants was measured using ELISA kits for OVA-induced IL-4, IL-5, IL-13, and IFN-γ. All data are presented as the mean and standard error from a minimum of three separate experiments and were compared using analysis of variance (ANOVA).

### 4.10. Quantitative Real-Time PCR (qRT-PCR)

Total RNA was isolated from the lung using RNA TRIzol reagent (Thermo Fisher Scientific, Waltham, MA, USA) according to the manufacturer’s instructions, and cDNA was synthesized from an equal amount of RNA (3 µg per reaction) using the First Strand cDNA Synthesis Kit (Amersham Pharmacia, Piscataway, NJ, USA). qRT-PCR was performed using a 7500 Fast Real-Time PCR system (Applied Biosystems, Foster City, CA, USA). The primer and probe sequences are shown in Table 3. 

*GAPDH* was used as the control for normalization. The analysis was performed using SYBR Green PCR Mastermix (Applied Biosystems). The following PCR thermal conditions were used: 2 min at 50 °C, and 10 min at 94 °C, followed by 35 cycles of 1 min at 94 °C, 1 min at 60 °C, and 1 min at 72 °C. The relative quantification (RQ) of target gene expression was performed using the comparative cycle threshold (CT) method. As previously described [61,63], qRT-PCR analyses were performed in triplicate and in accordance with the manufacturer’s protocols. 

### 4.11. Immunohistofluorescent (IHF) Staining

Immunohistochemical staining was performed as previously described with minor modifications [64]. Briefly, the lung tissues were frozen at −20 °C, and the frozen sections were randomly chosen and sliced in 20 µm thicknesses using a cryostat microtome (CM 3050S, Leica Microsystems, Wetzlar, Germany). Lung tissue sections (20 µm) were fixed with 4% sucrose and 4% paraformaldehyde in PBS at 20–25 °C for 40 min, permeabilized with 0.5% Nonidet P-40 in PBS, and blocked with 2.5% horse serum and 2.5% bovine serum albumin for 16 h. Double immunofluorescence staining was performed by incubating lung tissue sections with antibodies against GATA-3 (Santa Cruz Biotechnology, CA, USA), RORγt (BD Biosciences) and Foxp3 (Abcam, Cambridge, U.K.) overnight at 4 °C. Subsequently, a fluorescein-conjugated secondary antibody was added and incubated for 2 h, and nuclear staining was performed using Hoechst stain. Sections were observed using an Eclipse Ti-E inverted fluorescent microscope (Nikon Instruments Inc., Mississauga, ON, Canada).

### 4.12. In Vitro Experiments with EL-4 and LA-4 Cells

EL-4 T and LA-4 cells were plated in 6-well plates at a density of 5 × 10^5^ cells/mL. EL-4 cells were stimulated with PMA (1 ng/mL) and OP (5 μM) for 1 h. After pretreatment, CP (100 or 200 μg/mL) and OA (100 or 300 μM) were added to each plate. LA-4 cells were treated with vehicle and LPS (500 ng/mL) for 30 min prior to the addition of CP (100 or 300 μg/mL) and OA (100 or 300 μM). Then, EL-4 and LA-4 cells were cultured in IMDM (Iscove’s Modified Dulbecco’s Media) supplemented with 10% FBS, and 1% Gibco^®^ antibiotic antimycotic (containing 100 μg/mL streptomycin, 100 units/mL penicillin, and 0.25 μg/mL amphotericin B) in an incubator at 37 °C and 5% CO_2_ for 6 h. Total RNA (2 μg) was isolated using an Easy Blue kit (Intron Biotechnology, Inc., Seoul, Korea). Total RNA was reverse-transcribed into cDNA using the ReverTraAce cDNA Synthesis Kit (Toyobo, Osaka, Japan). Relative gene expression was quantified using the 7500 Fast Real-Time PCR system (Applied Biosystems). The protocol for total RNA extraction, cDNA synthesis, and qRT-PCR analysis is described in the qRT-PCR method section. 

### 4.13. Statistical Analysis

The results were analyzed using ANOVA or an unpaired Student’s *t*-test followed by Dunnett’s multiple comparison test (SPSS analysis software, version 14.0). Statistical significance is expressed in the figures and tables to emphasize the significance of the differences between the normal, dexamethasone, C-, OA, and OVA-induced control groups. The values are expressed as the mean ± standard error of the mean, and all test groups were compared against a control group. Values were considered statistically significant at *p* values of < 0.05 (*), < 0.01 (**), or < 0.001 (***) for the experimental groups versus the OVA control group comparisons and at *p* values of < 0.05 (^#^), < 0.01 (^##^), or < 0.001 (^###^) for the OVA-induced control group versus the normal group comparisons.

## 5. Conclusions

In summary, this study is the first to provide experimental evidence of the effects of ethanol extracts of CP and OA in a mouse model of OVA-induced asthma. CP and OA were shown to significantly inhibit airway and lung inflammation, mucus secretion, lymphocytes, neutrophil and eosinophil infiltration, AHR, and inflammatory mediators such as IL-5, IL-13, CCR3, MUC5AC, and COX-2 in the mouse model of asthma. CP and OA suppressed GATA-3 and RORγt expression and enhanced Foxp3 expression in lung tissues of the mice. The findings suggest that CP and OA may restore the balance of Treg/Th17 responses by suppressing GATA-3 and RORγt expression and promoting Foxp3 expression. 

## Figures and Tables

**Figure 1 molecules-26-01854-f001:**
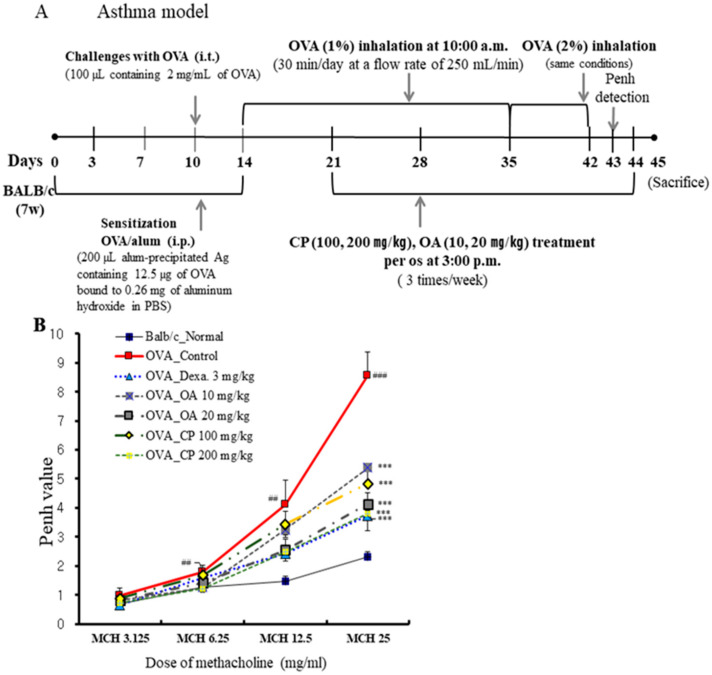
Schematic protocol illustration of ovalbumin (OVA)-induced mouse model of asthma (**A**). The suppressive effects of Cicadae Periostracum (CP) on airway hyperresponsiveness (AHR) in OVA-challenged mice were measured as enhanced pause (Penh) using non-invasive whole-body plethysmography (**B**). The effects of CP on total bronchoalveolar lavage fluid (BALF) cells (**C**), total lung cells (**D**), and neutrophils/eosinophils in BALF (**E**). BALF cytospin (400× images of cytospin slide) in the OVA-induced mouse model of allergic asthma (**F**). Results are expressed as mean ± standard error of the mean (SEM; N = 6). * *p* < 0.05, ** *p* < 0.01, and *** *p* < 0.001 for the OVA control group versus the experimental group comparisons; ^#^
*p* < 0.05, ^##^
*p* < 0.01, and ^###^
*p* < 0.001 for the OVA control group versus the normal group comparison. Normal, Normal BALB/c mice; Control, OVA inhalation plus vehicle; Dexa., OVA inhalation plus dexamethasone, 3 mg/kg; OA, OVA inhalation plus OA, 10 and 20 mg/kg; CP, OVA inhalation plus CP (100 or 200 mg/kg).

**Figure 2 molecules-26-01854-f002:**
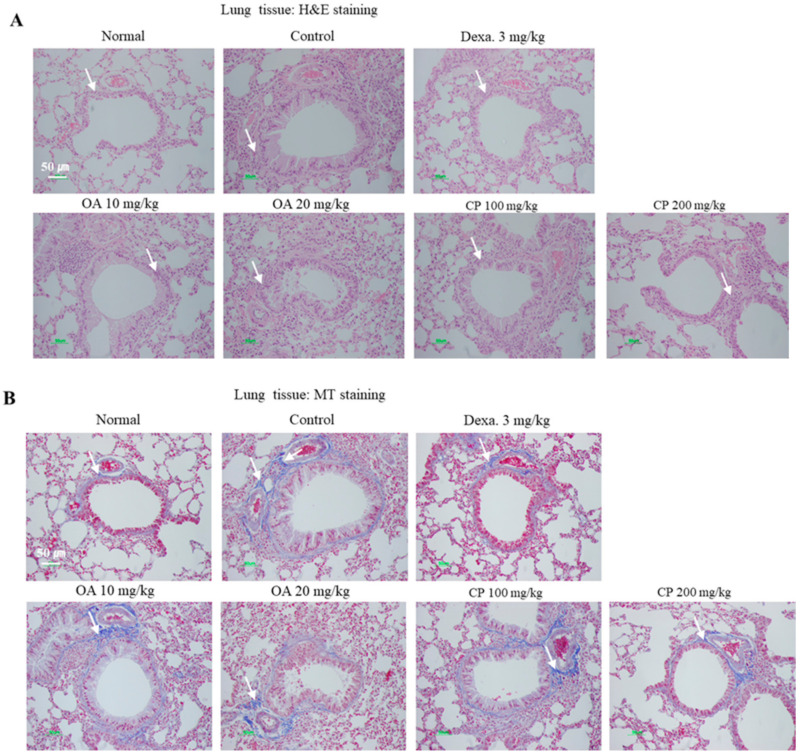
Histological examination of lung tissues. The lung sections were obtained from phosphate-buffered saline (PBS)- or OVA-challenged mice treated with Dexa. OA, and CP. Sections were stained with hematoxylin and eosin (H&E) for the analysis of the level of inflammation and lung tissue morphology (**A**). Sections were stained with Masson’s trichrome (M-T) for the quantitative analysis of the severity of collagen deposition and fibrosis (**B**). Sections were also stained with periodic acid-Schiff (PAS) or Alcian Blue PAS (AB/PAS) to determine the presence of goblet cells and mucus secretion (**C**, **D**). All of the randomly selected histological images were scored as the mean of inflammatory index scores (**E**). Results are expressed as mean ± SEM (N = 6). * *p* < 0.05, ** *p* < 0.01, and *** *p* < 0.001 for the OVA control group versus the experimental group comparisons. ^#^
*p* < 0.05, ^##^
*p* < 0.01, and ^###^
*p* < 0.001 for the OVA control group versus the normal group comparison. Normal, Normal BALB/c mice; Control, OVA inhalation plus vehicle; Dexa., OVA inhalation plus dexamethasone, 3 mg/kg; OA, OVA inhalation plus OA, 10 or 20 mg/kg; CP, OVA inhalation plus CP (100 or 200 mg/kg).

**Figure 3 molecules-26-01854-f003:**
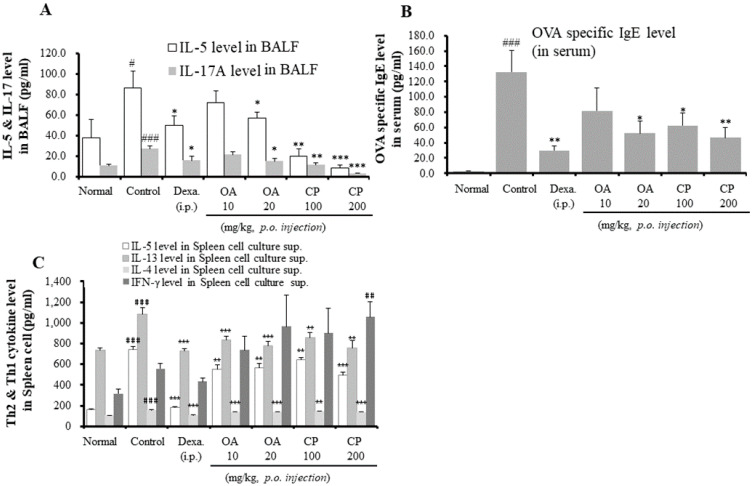
The effects of OA, CP, and Dexa on Th2 cytokine (IL-5) and IL-17A (**A**) in BALF; the levels of OVA specific IgE (**B**) in serum; and Th2/Th1 cytokines (IL-4, IL-5, IL-13, and interferon (IFN)-γ) (**C**). Each point represents the mean ± SEM values for 6 mice. * *p* < 0.05, ** *p* < 0.01, and *** *p* < 0.001 for the OVA control group versus the experimental group comparisons. ^#^
*p* < 0.05, ^##^
*p* < 0.01, and ^###^
*p* < 0.001 for the OVA control group versus the normal group comparison. Normal, Normal BALB/c mice; Control, OVA inhalation plus vehicle; Dexa., OVA inhalation plus dexamethasone, 3 mg/kg; OA, OVA inhalation plus OA, 10 or 20 mg/kg; CP, OVA inhalation plus CP (100 or 200 mg/kg).

**Figure 4 molecules-26-01854-f004:**
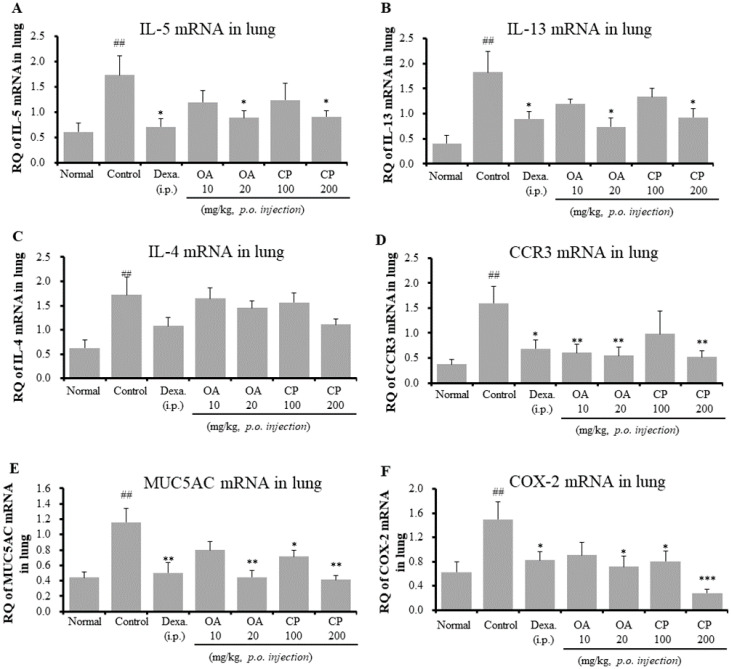
Effects of OA, CP, and Dexa on IL-5, IL-13, IL-4, CCR3, MUC5AC, and COX mRNA expression in the lung tissue in OVA-induced asthmatic model mice. IL-5 mRNA (**A**), IL-13 mRNA (**B**), IL-4 mRNA (**C**), CCR3 mRNA (**D**), MUC5AC mRNA (**E**), and COX-2 mRNA (**F**). Samples were analyzed by qRT-PCR. The results are expressed as relative quantification (RQ) to control. * *p* < 0.05, ** *p* < 0.01, and *** *p* < 0.001 for the OVA control group versus the experimental group comparisons. ^#^
*p* < 0.05, ^##^
*p* < 0.01, and ^###^
*p* < 0.001 for the OVA control group versus the normal group comparison.

**Figure 5 molecules-26-01854-f005:**
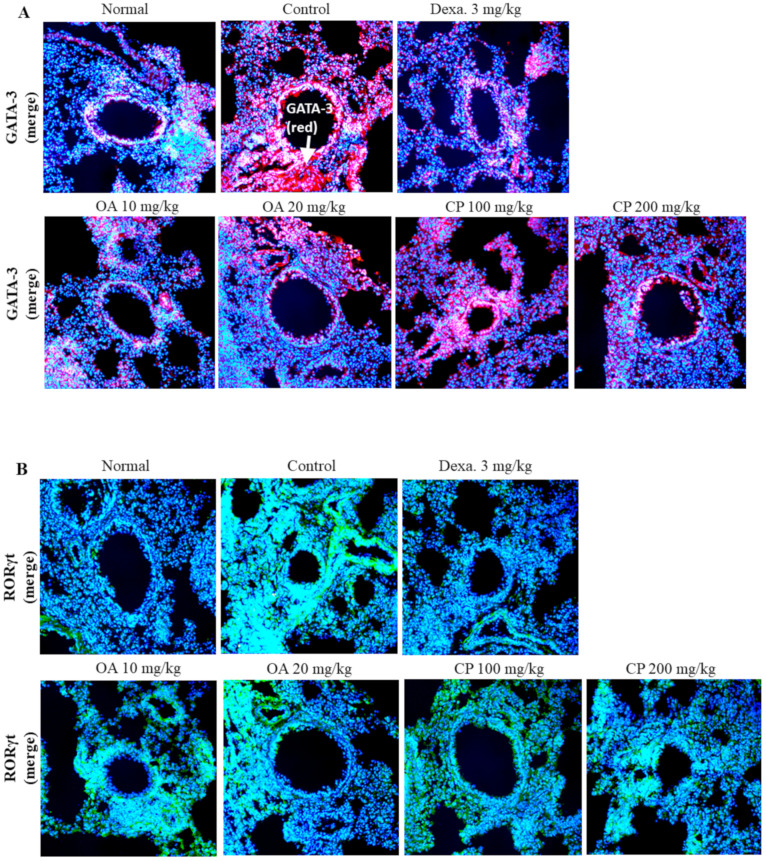
Suppressive effect of CP, OA, and Dexa. on GATA-3, RORγt and Foxp3 transcription factor using immunofluorescence analysis in lung of asthmatic model mice. The expressions of GATA-3 (**A**), RORγt (**B**), and Foxp3 (**C**) transcription factors in lung tissues were determined by immunofluorescence staining. The mean of fluorescence intensity of GATA-3, RORγt, and Foxp3 (**D**) quantified from images was obtained from 3 independent experiments using Image J software. The statistical significance of differences between control and treatment groups was assessed by ANOVA and Dunnett’s multiple comparison test. Each point represents the mean ± SEM values for 6 mice. ° Indicates a significant difference for the normal group versus the OVA control group (° *p* < 0.05 and °° *p* < 0.01); * in-dicates a significant difference (decrease) for the OVA control group versus the experimental group (* *p* < 0.05 and ** *p* < 0.01); ^#^ indicates a significant difference (increase) for the OVA control group versus the experimental group (^#^
*p* < 0.05 and ^##^
*p* < 0.01). Normal, Normal BALB/c mice; Control: OVA inhalation plus vehicle; Dexa., OVA inhalation plus dexamethasone, 3 mg/kg; OA, OVA inhalation plus OA, 10 or 20 mg/kg; CP, OVA inhalation plus CP (100 or 200 mg/kg). White arrows indicate each protein expressions.

**Figure 6 molecules-26-01854-f006:**
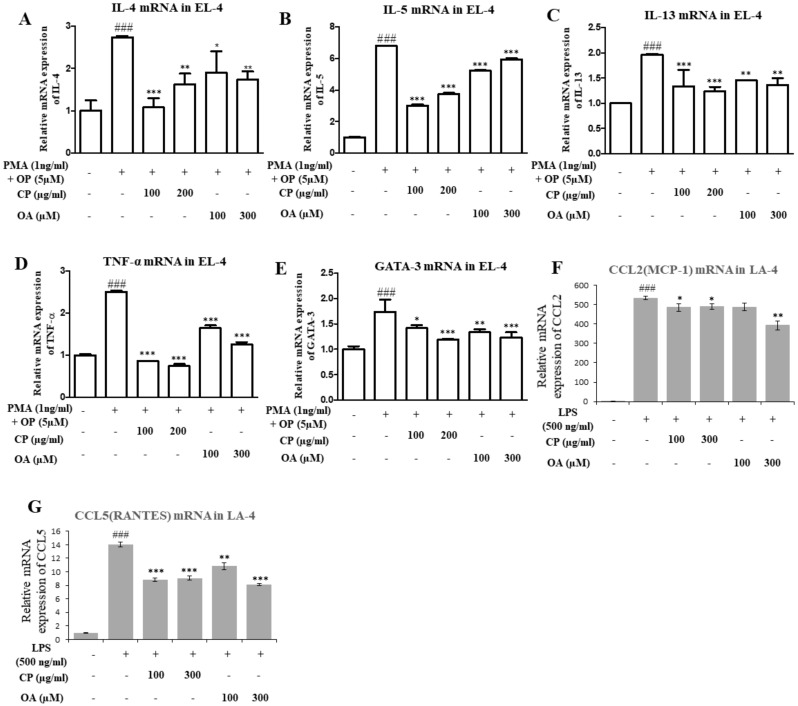
Effect of OA, CP on phorbol 12-myristate 13-acetate (PMA)- and 4-tertoctylphenol (OP)-induced cytokine gene expression of IL-4 (**A**), IL-5 (**B**), IL-13 (**C**), TNF-α (**D**), and GATA-3 (**E**) in EL-4 T cells; and CCL2 (**F**) and CCL5 (**G**) in LA-4 cells. EL-4 cells were stimulated with phorbol 12-myristate 13-acetate (PMA, 1 ng/mL), and 4-tertoctylphenol (OP, 5 μM) for 1 h. After pretreatment, CP (100 or 200 μg/mL) and OA (100 or 300 μM) were added to each plate. LA-4 cells were treated with vehicle, lipopolysaccharides (LPS, 500 ng/mL), for 30 min prior to the addition of CP (100 or 300 μg/mL) and OA (100 or 300 μM). Then, these cells (EL-4, LA-4) were cultured in an incubator at 37 °C and 5% CO_2_ for 6 h. mRNA expression levels of these cytokines in EL-4 and LA-4 cells were measured using quantitative real-time PCR. The data shown represent the mean ± SEM of three independent experiments. The statistical significance of differences between control and treatment groups were assessed by ANOVA and Dunnett’s multiple comparison test. (^#^
*p* < 0.05 and ^###^
*p* < 0.001 v. the normal control group; * *p* < 0.05, ** *p* < 0.01, and *** *p* < 0.001 vs. PMA- and OP-treated or LPS-treated group).

**Table 1 molecules-26-01854-t001:** Composition of fatty acids and total fatty acids in *C. pustulata* (CP) (g/100 g).

Abbreviation	Compounds (Fatty Acid)	Formula	Result (g/100 g)
C6:0	Caproic acid	CH_3_(CH_2_)_4_COOH	0.002
C8:0	Caprylic acid	CH_3_(CH_2_)_6_COOH	0.002
C12:0	Lauric acid	CH_3_(CH_2_)_10_COOH	0.002
C14:0	Myristic acid	CH_3_(CH_2_)_12_COOH	0.015
C15:0	Pentadecanoic acid	CH_3_(CH_2_)_13_COOH	0.004
C16:0	Palmitic acid	CH_3_(CH_2_)_14_COOH	0.187
C16:1	Palmitoleic acid	CH3(CH2)5CH=CH(CH2)7COOH	0.015
C17:0	Heptadecanoic acid	CH_3_(CH_2_)_15_COOH	0.006
C18:0	Stearic acid	CH_3_(CH_2_)_16_COOH	0.093
C18:1 (trans) C18:1 (cis)	Oleic acid	CH3(CH2)7CH=CH(CH2)7COOH	0.018 (trans) 0.156 (cis)
C18:2 (trans) C18:2 (cis)	Linoleic acid	CH3(CH2)4CH=CHCH2CH=CH(CH2)7COOH	0.006 (trans) 0.057 (cis)
C20:0	Arachidic acid	CH_3_(CH_2_)_18_COOH	0.006
C20:1	Gondoic acid	CH_3_(CH_2_)_7_CH=CH(CH_2_)_9_COOH	0.008
C22:0	Behenic acid	CH_3_(CH_2_)_20_COOH	0.004
C22:1n-9	Erucic acid	CH₃(CH₂)₇CH=CH(CH₂)₁₁COOH	0.025
C22:2	Docosadienoic acid	C_21_H_39_COOH	0.009
C24:0	Lignoceric acid	C₂₃H₄₇COOH	0.003
Total fatty acids			0.62

**Table 2 molecules-26-01854-t002:** Suppressive effect of OA, CP, and Dexa on CD11b^+^Gr-1^+^(high) neutrophils and CD11b^+^SiglecF^+^ eosinophils in the BALF and lung of OVA-induced asthmatic model mice.

Cell Phenotypes in BALF and Lung		OVA-Induced Airway Inflammation Murine Model (Absolute No.)						
		Normal	Control	Dexa. 3 mg/kg	OA 10 mg/kg	OA 20 mg/kg	CP 100 mg/kg	CP 200 mg/kg
CD11b^+^Gr-1^+^ (×10^5^ cells)	BALF	0.96 ± 0.13	1.72 ± 0.70	0.39 ± 0.12 *	0.27 ± 0.12 *	0.48 ± 0.08 *	0.62 ± 0.13	0.21 ± 0.04 *
CD11b^+^SiglecF^+^(×10^5^ cells)		1.26 ± 0.20	42.8 ± 3.11 ^###^	7.52 ± 0.73 ***	22.36 ± 3.80 ***	12.3 ± 2.21 **	26.7 ± 3.63 **	9.34 ± 1.02 ***
CD11b^+^Gr-1^+^ (×10^5^ cells)	Lung	5.11 ± 0.36	19.31 ± 1.10 ^###^	5.30 ± 0.60 ***	7.88 ± 0.70 ***	5.87 ± 0.19 ***	8.26 ± 0.69 ***	5.50 ± 0.47 ***
CD11b^+^SiglecF^+^(×10^5^ cells)		6.65 ± 0.30	39.0 ± 2.90 ^###^	12.61 ± 1.28 ***	22.49 ± 1.40 ***	14.99 ± 1.24 ***	16.50 ± 1.18 ***	12.53 ± 1.42 ***

Samples were measured on a flow cytometer and analyzed by two-color flow cytometry using a FACSCalibur device and CellQuest software (BD Biosciences, Mountain View, CA, USA). Absolute cell number of CD11b+Gr-1+(high) neutrophils and CD11b^+^/SiglecF^+^ eosinophils present in the BALF and lung. The statistical significance of differences between control and treatment groups was assessed by ANOVA and Dunnett’s multiple comparison test. Each point represents the mean ± SEM values for 6 mice. * *p* < 0.05, ** *p* < 0.01, and *** *p* < 0.00, for the OVA control group versus the experimental group comparisons. ^#^
*p* < 0.05, ^##^
*p* < 0.01, and ^###^
*p* < 0.001 for the OVA control group versus the normal group comparison. Normal, Normal BALB/c mice; Control, OVA inhalation plus vehicle; Dexa., OVA inhalation plus dexamethasone, 3 mg/kg; OA, OVA inhalation plus OA, 10 or 20 mg/kg; CP, OVA inhalation plus CP (100 or 200 mg/kg).

**Table 3 molecules-26-01854-t003:** Primers and probe sequence used in real-time PCR analysis.

Gene	Primer	Oligonucleotide Sequence (5′–3′)
GAPDH	F	5′-CAATGAATACGGCTACAGCAAC-3′
	R	5′-AGGGAGATGCTCAGTGTTGG-3′
IL-5	F	5′-AGCACAGTGGTGAAAGAGACCTT-3′
	R	5′-TCCAATGCATAGCTGGTGATTT-3′
IL-4	F	5′-GGATGTAACGACAGCCCTCT-3′
	R	5′-GTGTTCCTTGTTGCCGTAAG-3′
IL-13	F	5′-CAGTTGCAATGCCATCCACA-3′
	R	5′-AGCCACATCCGAGGCCTTT-3′
CCR3	F	5′-CCCGAACTGTGACTTTTGCT-3′
	R	5′-CCTCTGGATAGCGAGGACTG-3′
MUC5AC	F	5′-AGAATATCTTTCAGGACCCCTGCT-3′
	R	5′-ACACCAGTGCTGAGCATACTTTT-3′
COX-2	F	5′-TCTCAGCACCCACCCGCTCA-3′
	R	5′-GCCCCGTAGACCCTGCTCGA-3′
CCL-2	F	5′-CAGCAGGTGTCCCAAAGAAG-3′
	R	5′-TGTGGAAAAGGTAGTGGATGC-3′
CCL-5	F	5′-ATATGGCTCGGACACCACTC-3′
	R	5′-TTCTTCGAGTGACAAACACG-3′
TNF-ɑ	F	5′-ATGAGCACAGAAAGCATGAT-3′
	R	5′-CACACCGACCTTCACCATTTT-3′
GATA-3	F	5′-CTCCTTTTTGCTCTCCTTTTC-3′
	R	5′-AAGAGATGAGGACTGGAGTG-3′
IL-4 (In vitro)	F	5′-GGCTTCCAAGGTGCTTC-3′
	R	5′-CAGGCATCGAAAAGCC-3′
IL-5 (In vitro)	F	5′-TTCCTGAAGGCTGAGGTTAC-3′
	R	5′-AACATGCACAAAGCCTTGGG-3′
IL-13 (In vitro)	F	5′-AGAGGATATTGCATGGCCTCTG-3′
	R	5′-TGCTTTGTGTAGCTGAGCAG-3′

## Data Availability

Data is contained within the article or Appendix A.

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
