# Peer review of "Cryptotympana pustulata Extract and Its Main Active Component, Oleic Acid, Inhibit Ovalbumin-Induced Allergic Airway Inflammation through Inhibition of Th2/GATA-3 and Interleukin-17/RORγt Signaling Pathways in Asthmatic Mice"

_molecules, 2021, doi:10.3390/molecules26071854_

Round 1

Reviewer 1 Report

The study of “Cryptotympana pustulata Extract and Its Main Active Component Oleic Acid Inhibit Ovalbumin-Induced Allergic Airway Inflammation through Inhibition of the Th2/GATA-3 and Interleukin-17/RORγt Signaling Pathways in Asthmatic Mic” has good and clear explanation in Introduction, M&M, Results and Discussion. However, reviewer has some suggestions and questions.

Major comments:

  1. According the literatures, the bioactive compounds of CP may be N-acetyldopamine. Why didn’t author detect N-acetyldopamine derivatives in this manuscript?

Ref: 1. Thapa P, Gu Y, Kil YS, Baek SC, Kim KH, Han AR, Seo EK, Choi H, Chang JH, Nam JW. , N-Acetyldopamine derivatives from Periostracum Cicadae and their regulatory activities on Th1 and Th17 cell differentiation, Bioorg Chem. 2020 Sep;102:104095. doi: 10.1016/j.bioorg.2020.104095.

2. Xu MZ, Lee WS, Han JM, Oh HW, Park DS, Tian GR, Jeong TS, Park HY., Antioxidant and anti-inflammatory activities of N-acetyldopamine dimers from Periostracum Cicadae, Bioorg Med Chem.2006 Dec 1;14(23):7826-34.doi: 10.1016/j.bmc.2006.07.063.

  1. The content of palmitic acid (0.187g/100g of CP) is more than oleic acid (cis and trans form: 0.174 g/100g of CP). Could the author explain the effect of palmitic acid for OVA-induced asthma?
  2. According ref 12, OA was positively associated with adult asthma. Could the author explain why the results of OA in animal experiments is different from clinic report?
  3. Since the content of IgG, for example IgG1 and IgG2, is another marker for asthma, did the author detect the concentration of IgG in the serum?

Minor comments:

  1. The words resolution of the Figures, especially Y-axis, is not enough clear.please checked that.
  2. IL-17 in Fig 5 was IL-17A, and IFN gamma was IFN-g. please checked and corrected it.
  3. There were seven bar for Fig 4E, but there was no explain for seventh bar. Please checked and explained that. And the spacing between groups can be adjusted larger, so that readers can clearly view the relevant data.

Author Response

Firstly, Reviewer #1 comments1: The study of “Cryptotympana pustulata Extract and Its Main Active Component Oleic Acid Inhibit Ovalbumin-Induced Allergic Airway Inflammation through Inhibition of the Th2/GATA-3 and Interleukin-17/RORγt Signaling Pathways in Asthmatic Mic” has good and clear explanation in Introduction, M&M, Results and Discussion. However, reviewer has some suggestions and questions.

Major comments:

  1. According the literatures, the bioactive compounds of CP may be N-acetyldopamine. Why didn’t author detect N-acetyldopamine derivatives in this manuscript?

Ref: 1. Thapa P, Gu Y, Kil YS, Baek SC, Kim KH, Han AR, Seo EK, Choi H, Chang JH, Nam JW. , N-Acetyldopamine derivatives from Periostracum Cicadae and their regulatory activities on Th1 and Th17 cell differentiation, Bioorg Chem. 2020 Sep;102:104095. doi: 10.1016/j.bioorg.2020.104095.

  1. Xu MZ, Lee WS, Han JM, Oh HW, Park DS, Tian GR, Jeong TS, Park HY., Antioxidant and anti-inflammatory activities of N-acetyldopamine dimers from Periostracum Cicadae, Bioorg Med Chem.2006 Dec 1;14(23):7826-34.doi: 10.1016/j.bmc.2006.07.063.

RESPONSE: ● First of all, thank you for your kind meaningful and helpful comment. (briefly added in the introduction)  

Insects (including) can be a more efficient source of fatty acids, and protein. Also, insect oils (fat) are a kind of nutrient substances with several physiological and biological activities and functions. It has a high value of research, development and utilization no matter whatever be the quantity or quality.(

Liu XG, Ju XR, Wang HF, et al. Insect oil and its nourishment appraisement. Journal of the Chinese Cereals and Oils Association. 2003;6:11‐13)

Recent investigations have shown that there are 24 kinds of trace elements, a lot of amino acids, fatty acids, protein, chitin, and acetyl-dopamine dimers in the Cicada slough. In many research studies, fatty acids (especially oleic acid) exhibit anti-inflammatory activity and affect the biosynthesis of prostaglandins, which play an important role in allergic diseases. However, there are a few reports about N-acetyldopamine. Moreover, studies on the possibility of candidate substances for N-acetyldopamine are insufficient, and no meaningful evidence has been found regarding asthma, allergy and etc.

Therefore, we select fatty acids (oleic acid; most abundant fatty acid) as a representative ingredient of CP. As previously explained in the text (introduction), fatty acids were thought to be the key ingredients (bioactive component).

[Reference 9] Chem. Nat. Compd. 2015, 511147–1148.

N-acetyldopamine is a secondary carboxamide obtained by formal condensation of the carboxy group of acetic acid with the amino group of dopamine. It is a secondary carboxamide, a member of acetamides and a member of catechols. The content of N-acetyldopamine dimer A and N-acetyldopamine dimer B, two representative N-acetyldopamine dimers in CP, were 0.305∼0.794 mg/g. However, in vivo metabolism of these N-acetyldopamine oligomers is still unclear. Metabolism can affect a drug’s distribution, rate or route of excretion and production of new and possibly active or toxic metabolites. The absorbed prototypes and their metabolites of a drug might be the active constituents. However, the exposed components and metabolites of N-acetyldopamine oligomers of CP (NOCP) in vivo are still unknown.

  •  C. Cao, X.Y. Zhang, J.D. Xu, H. Shen, S.S. Zhou, H. Zhu, M. Kong, W. Zhang, G.R.Zhou, Y. He, Q. Mao, S.L. Li, Quality consistency evaluation on four origins of Cicadae Periostracum by ultra-performance liquid chromatography coupledwith quadrupole/time-of-flight mass spectrometry analysis, J. Pharm.Biomed. Anal. 179 (2020), 112974.
  • Journal of Pharmaceutical and Biomedical Analysis 192 (2021) 113665

Although it is not sufficient direct data and our results are not strongly indicated each compositions, we proved the significant(meaningful) evidence concerning them (between CP including oleic acid, major compound, and Th2 responses, GATA-3).   

Therapeutic activities and mechanisms of fatty acids, amino acids, N-acetyldopamine and their mixtures in asthma model and allergic disorders have not been fully investigated individually. Therefore, it must be considered and investigated in the future work as an another study.

Reviewer #1 comments2: The content of palmitic acid (0.187g/100g of CP) is more than oleic acid (cis and trans form: 0.174 g/100g of CP). Could the author explain the effect of palmitic acid for OVA-induced asthma?

RESPONSE: ● Thank you for your kind meaningful and helpful comment.

The content of oleic acid and palmitic acid is not thought to be much different. In many cases, oleic acid is known to have a more meaningful effect on allergic diseases including asthma than on palmitic acid. However, the role of palmitic acid in asthma are still unclear (in asthma model) or controversial as follows;

  • Palmitic acid was increased in OVA-induced asthma model compared with control group. Journal of Chromatography B 1063 (2017) 156–162
  • The administration of palmitic acid sensitize DCs resulting in augmented secretion of TH1/TH17-instructive cytokines upon pro-inflammatory stimulation. J. Immunol. 2016. 46: 2043–2053
  • In a meaningful and comparable result, Seo et al. reported that palmitic acid and methionine were the most common metabolites, as potential biomarkers in the plasma samples of OVA-induced asthmatic mice. Chromatogr. B Analyt. Technol. Biomed. Life Sci. 2017, 1063, 156–162.
  • There is an evidence about the potential usefulness of the mixture of fatty acids (palmitic, oleic, linoleic and linolenic acids) in cutaneous inflammatory and allergic disorders. Planta Med. 2005 Feb;71(2):126-9.
  • High intake of saturated fatty acids (palmitic acid) was associated with decreased risk of asthma in offspring at 5 y. (J Allergy Clin Immunol 2014;133:1255-64.)

As reviewer2’ s comment, the stimulatory effects of CP extract shown in the various experiments are due to the combined action of all the intact components: proteins, peptides; glycopeptides; lipids; glycides etc.

Herbal products contain plant, animal(insect) extracts, which are complex mixtures of various compounds. As with traditional drugs, it is necessary to validate their (individual ingredients and their mixture) efficacy and safety through preclinical and clinical studies.

Reviewer #1 comments3: According ref 12, OA was positively associated with adult asthma. Could the author explain why the results of OA in animal experiments is different from clinic report?

RESPONSE: Thank you for your kind meaningful and helpful comment.

We think that high, low intake (according to dose), metabolites in humans and animals, or different fatty acids mixture (or with other components, etc.) by various ratios (of each fatty acids) may induce different results in animal disease model or clinic report.

According ref. 12, OA was administered high concentration (high dietary intake of OA).

  • Oleic acid is supposed to present modulatory effects in a wide physiological functions, while some studies also suggest a beneficial effect on cancer, autoimmune and inflammatory diseases, besides its ability to facilitate wound healing. Although the OA role in immune responses are still controversial, the administration of olive oil containing diets may improve the immune response. Mini Rev Med Chem 2013 Feb;13(2):201-10.
  • Oleic acid-induced lung injury is a widely used model resembling the human disease. The oleic acid has been linked to metabolic and inflammatory diseases; focused on lung injury.

Mediators Inflamm. 2015;2015:260465.

In my view, it is thought that changes (conflicting results) will be reflected depending on the concentration (or combination with other components, metabolites) administered for each study.

  • It has long been hypothesized that decreasing the n-6/n-3 ratio (including C18/1(n-9)) could reduce the production of more proinflammatory mediators while increasing the formation of downstream metabolites that can serve to limit or resolve inflammation. In turn, these changes would result in improved asthma outcomes or would lower the risk for asthma incidence. (J Allergy Clin Immunol 2014;133:1255-64.)
  • The higher consumption of fatty acids and the high ratio of n-3 to n-6 PUFA may be associated with a lower prevalence of asthma. Nutrients 2019, 11, 2187

Asthma is a heterogenous disease, therefore various factors in the clinical characteristics of the study population should be taken into consideration. Additionally, metabolites in humans and animals, differences in experimental design, inclusion criteria and methodology may lead to the differences in the reported findings. All future studies should consider unified inclusion and exclusion criteria and methodology, and account for additional essential variables including other nutrients and microbiome composition. (clinic, animal differences)

Reviewer #1 comments4: Since the content of IgG, for example IgG1 and IgG2, is another marker for asthma, did the author detect the concentration of IgG in the serum?

RESPONSE: Thank you for your kind meaningful and helpful comment.

In our study, we focused on neutrophil, eosinophil infiltration, IgE, Th2 cytokines (IL-5, IL-13, IL-4), GATA-3, Th17, RORγT, Foxp3 in asthma model mice.

Unfortunately, we did not try evaluating the content of IgG as another marker for asthma in the serum. As reviewer’s meaningful and helpful comment, further investigation about the quantification of IgG in the serum (functional outcome of asthma model) should be accomplished in separate study including this factor in the future study.

Reviewer #1 Minor comments 1: The words resolution of the Figures, especially Y-axis, is not enough clear.please checked that.

RESPONSE: Thank you for your kind meaningful and helpful comment.

As reviewer’s comment we checked and revised the figures, especially Y-axis. (Fig.5, 6 etc.)

Some figures are presented in tabular form for better analysis (as reviewer2’s comment).

Reviewer #1 Minor comments 2: IL-17 in Fig 5 was IL-17A, and IFN gamma was IFN-g. please checked and corrected it.

RESPONSE: Thank you for your kind meaningful and helpful comment.

As reviewer’s comment we checked and revised the figures, especially IL-17A, IFN-γ.

Reviewer #1 Minor comments 3: There were seven bar for Fig 4E, but there was no explain for seventh bar. Please checked and explained that. And the spacing between groups can be adjusted larger, so that readers can clearly view the relevant data.

RESPONSE: Thank you for your kind meaningful and helpful comment.

Item “goblet” was hidden from the legend mark (legend box displayed small). – corrected (adjusted larger). Also, spacing between groups was adjusted larger as reviewer’ suggestion.

Reviewer 2 Report

In the manuscript "Cryptotympana pustulata Extract and Its Main Active Component Oleic Acid Inhibit Ovalbumin-Induced Allergic Airway Inflammation through Inhibition of the Th2/GATA-3 and Interleukin-17/RORγt Signaling Pathways in Asthmatic Mice"  the authors investigate the anti-inflammatory effect of Cryptotympana pustulata extract and Oleic Acid and their possible involvement in the treatment of asthma and airway inflammation.

The manuscript is of great interest to the scientific community and is well structured however the large number of figures and the presentation of the results in multiple forms make it difficult to understand.

Introduction:

The authors present in advance the results obtained in this section: page 2 line 48. The text should be revised to eliminate this presentation

Major comments:

Section 2.1 

Figure 1 and Table 1 should be deleted. The authors present the total amino acid profile of the ethanolic extract of CP. In the context of this work the result is not meaningful, none of the anti-inflammatory effects verified can be correlated or discussed with the total amino acid constitution. The stimulatory effects of CP extract shown in the various experiments are due to the combined action of all the intact components: proteins, peptides; glycopeptides; lipids; glycides etc.

Figure 2 should be deleted or presented as supplementary material since the results are perfectly presented by the table 2

Section 2.2

Figure 3B: The graph should be improved by using distinct colours on the various curves.

Section 2.4

Figure 5A and Figure 5B should be merged into one graph

Section 2.6

Figure 7: Figures A; B; C; D may be presented in supplementary material and the data in figures 7:E, F, G and H should be presented in tabular form for better analysis.

Section 2.7

Figure 8: The results of Figure B; D and F should be combined into one graph.

Discussion:

The authors should justify and discuss the choice of the quantities used in the various experiments (100 and 200 mg for the CP extracts and 10 and 20 mg of OA). Lines 461 to 463 indicate the existence of preliminary results on the safety and efficacy of the quantities chosen. These results should be presented.

Conclusions:

The statements contained in lines 692-697 should be reviewed. This type of statements can be present in the discussion of a paper but should not be included in the Conclusions as they can easily be used by the reviewer  to refuse the publication of the paper...

Author Response

Reviewer #2 comments1: 1. Introduction: The authors present in advance the results obtained in this section: page 2 line 48. The text should be revised to eliminate this presentation.

RESPONSE:  Thank you for your kind meaningful and helpful comment.

We revised to eliminate this presentation as reviewer’s comment.

Reviewer #2 comments: Section 2.1  Figure 1 and Table 1 should be deleted. The authors present the total amino acid profile of the ethanolic extract of CP. In the context of this work the result is not meaningful, none of the anti-inflammatory effects verified can be correlated or discussed with the total amino acid constitution. The stimulatory effects of CP extract shown in the various experiments are due to the combined action of all the intact components: proteins, peptides; glycopeptides; lipids; glycides etc.

Figure 2 should be deleted or presented as supplementary material since the results are perfectly presented by the table 2

RESPONSE:  Thank you for your kind meaningful and helpful comment.

We revised to eliminate Figure1, table 1, figure2 as reviewer’s comment. (all figure, table numbers were changed)

Section 2.2 Figure 3B: The graph should be improved by using distinct colours on the various curves.

The graph was improved by using distinct colours on the various curves as reviewer’s comment

Section 2.4 Figure 5A and Figure 5B should be merged into one graph

Figure 5A and Figure 5B was merged into one graph as reviewer’s comment

Section 2.6 Figure 7: Figures A; B; C; D may be presented in supplementary material and the data in figures 7:E, F, G and H should be presented in tabular form for better analysis.

Figures A; B; C; D presented proportion of each cell subtypes (representative figure). Main results was absolute number of each cell types. As reviewer’s suggestion, the large number of figures and the presentation of the results in multiple forms make it difficult to understand. Therefore, we removed the Fig.7 A,B,C,D, and results was presented in tabular form for better analysis (table 2).

Section 2.7 Figure 8: The results of Figure B; D and F should be combined into one graph.

Figure 8: The results of Figure B; D and F was combined into one graph as reviewer’s comment.  Also, we replaced the staining image only with merged (with Hoechst) as reviewer3’s comment.

Discussion:The authors should justify and discuss the choice of the quantities used in the various experiments (100 and 200 mg for the CP extracts and 10 and 20 mg of OA). Lines 461 to 463 indicate the existence of preliminary results on the safety and efficacy of the quantities chosen. These results should be presented.

RESPONSE:  Thank you for your kind meaningful and helpful comment.

CP is a commonly consumed insect that is also used as an ingredient in the formulation of Traditional medicine in China, Korea etc. As the prevailing food cultures, CP (edible insects) could be dated back to ancient Asia. Cicadae Periostracum is a commonly used crude drug in traditional medicine in Asia.

We used (selected) two representative doses (high and low, selected minimal doses as effective and safe range) of the tested materials based on preliminary reports listed below. Moreover, we evaluated hematologic toxicity of each groups by hematological analysis (There were no significant hematologic toxicities; supplementary data Table 1), and investigated by western blot, cell viability assay for supporting data (added in supplementary material, Figure S1, S2). Additional references are added in the text.

  • Abreu, P.; Pinheiro, C.H.; Vitzel, K.F.; Vasconcelos, D.A.; Torres, R.P.; Fortes, M.S.; Marzuca-Nassr, G.N.; Mancini-Filho, J.; Hirabara, S.M.;, Curi R. Contractile function recovery in severely injured gastrocnemius muscle of rats treated with either oleic or linoleic acid. Physiol.. 2016, 101, 1392-1405.
  • García-Cerro, S.; Rueda, N.; Vidal, V.; Puente, A.; Campa, V.; Lantigua, S.; Narcís, O.; Velasco, A.; Bartesaghi, R.; Martínez-Cué, C.; et al. Prenatal Administration of Oleic Acid or Linolenic Acid Reduces Neuromorphological and Cognitive Alterations in Ts65dn Down Syndrome Mice. Nutr. 2020, 150, 1631-1643.
  • Meng, Y.; Zhang, J.; Yuan, C.; Zhang, F.; Fu, Q.; Su, H.; Zhu, X.; Wang, L.; Gao, P.; Shu, G.; et al. Oleic acid stimulates HC11 mammary epithelial cells proliferation and mammary gland development in peripubertal mice through activation of CD36-Ca(2+) and PI3K/Akt signaling pathway. 2018, 9, 12982-12994.
  • Wang, J.; Zhang, Y.; Fang, Z.; Sun, L.; Wang, Y.; Liu, Y.; Xu, D.; Nie, F.; Gooneratne, R. Oleic Acid Alleviates Cadmium-Induced Oxidative Damage in Rat by Its Radicals Scavenging Activity. Trace Elem. Res. 2019, 190, 95-100.

  • Lim, H.S.; Kim, J.S.; Moon, B.C.; Choi, G.; Ryu, S.M.; Lee, J.; Ang, M.J.; Jeon, M.; Moon, C.; Park, G. Cicadidae Periostracum, the Cast-Off Skin of Cicada, Protects Dopaminergic Neurons in a Model of Parkinson's Disease. Oxid Med Cell Longev. 2019, 2019, 5797512.
  • Shin, T.Y.; Park, J.H.; Kim, H.M.Effect of Cryptotympana atrata extract on compound 48/80-induced anaphylactic reactions. J. Ethnopharmacol. 1999, 66, 319-325
  • Yang, L.; Wang, Y.; Nuerbiye, A.; Cheng, P.; Wang, J.H.; Kasimu, R.; Li, H. Effects of Periostracum Cicadae on Cytokines and Apoptosis Regulatory Proteins in an IgA Nephropathy Rat Int. J. Mol. Sci. 2018, 19, 1599.
  • Yen, H.R.; Liang, K.L.; Huang, T.P.; Fan, J.Y.; Chang, T.T.; Sun, M.F.Characteristics of traditional Chinese medicine use for children with allergic rhinitis: a nationwide population-based study. J. Pediatr. Otorhinolaryngol. 2015, 79, 591-597.

Etc.

Conclusions:The statements contained in lines 692-697 should be reviewed. This type of statements can be present in the discussion of a paper but should not be included in the Conclusions as they can easily be used by the reviewer  to refuse the publication of the paper...

RESPONSE:  Thank you for your kind meaningful and helpful comment.

We agree with the reviewer’s comment and we think it's a repetitive expression (it was already expressed in the discussion section).

Therefore, we revised to eliminate above statements as reviewer’s comment.

Reviewer 3 Report

The authors demonstrate that cicadae Periostracum may have a potential for the treatment of asthma via inhibition of 30 the GATA-3/Th2 and IL-17/RORγt signaling pathways.

The study is well performed. However, the flow analysis has to be analyzed with the proper separation of cells. The immunofluorescence staining of GATA-3, FOXP3, RORgT cells looks too high, no specific cells shown/visible. The staining must be replaced only with merged cells with DAPI, remove DAPI staining show only DAPI merged cells.

Author Response

Reviewer #3 comments: The study is well performed. However, the flow analysis has to be analyzed with the proper separation of cells. The immunofluorescence staining of GATA-3, FOXP3, RORgT cells looks too high, no specific cells shown/visible. The staining must be replaced only with merged cells with DAPI, remove DAPI staining show only DAPI merged cells.

RESPONSE:  Thank you for your kind meaningful and helpful comment.

We agree with the reviewer’s comment in principle. We replaced the staining image only with merged (with Hoechst). (revised) And the results of Figure B; D and F (relative graph) were combined into one graph as reviewer2’s comment.

Even if it's not enough, we think that the difference of their degree of staining can be discriminable through each comparison in this result. 

Although other reviewer suggested moderate english change, the text was revised with correct language and better descriptions (or checked again) as possible. (English correction has been made MDPI provides an English editing service, https://www.mdpi.com/authors/english. English editing article ID; 28129

Round 2

Reviewer 2 Report

The present manuscript responds to all the comments made by the reviewers. 
I suggest a minor correction in  "Section 2.1 Chemical Profiles of Amino Acids and Fatty Acids in CP Ethanol Extract and Fatty Acids" , page 3 line 97, as the amino acid profile is no longer part of the results presented.

Author Response

Reviewer #2 comments1: The present manuscript responds to all the comments made by the reviewers. 
I suggest a minor correction in  "Section 2.1 Chemical Profiles of Amino Acids and Fatty Acids in CP Ethanol Extract and Fatty Acids" , page 3 line 97, as the amino acid profile is no longer part of the results presented.
RESPONSE:  Thank you for your kind meaningful and helpful comment. 
We revised to eliminate this presentation (amino acid) as reviewer’s comment. as follows;
"2.1. Chemical Profiles of Fatty Acids in CP Ethanol Extract"